# Appropriateness of Care for Common Childhood Infections at Low-Level Private Health Facilities in a Rural District in Western Uganda

**DOI:** 10.3390/ijerph18157742

**Published:** 2021-07-21

**Authors:** Juliet Mwanga-Amumpaire, Tobias Alfvén, Celestino Obua, Karin Källander, Richard Migisha, Cecilia Stålsby Lundborg, Grace Ndeezi, Joan Nakayaga Kalyango

**Affiliations:** 1Faculty of Medicine, Mbarara University of Science and Technology, Mbarara P.O. Box 1410, Uganda; celestino.obua@must.ac.ug (C.O.); rmigisha@must.ac.ug (R.M.); 2Clinical Epidemiology Unit, College of Health Sciences, Makerere University, Kampala P.O. Box 7062, Uganda; nakayaga2001@yahoo.com; 3Department of Global Public Health, Karolinska Institutet, 171 77 Solna, Sweden; tobias.alfven@ki.se (T.A.); karin.kallander@ki.se (K.K.); cecilia.stalsby.lundborg@ki.se (C.S.L.); 4Sachs’ Children and Youth Hospital, 118 83 Stockholm, Sweden; 5Programme Division, Health Section, UNICEF, New York, NY 10017, USA; 6Department of Pediatrics and Child Health, College of Health Sciences, Makerere University, Kampala P.O. Box 7062, Uganda; gndeezi@gmail.com; 7Department of Pharmacy, College of Health Sciences, Makerere University, Kampala P.O. Box 7062, Uganda

**Keywords:** appropriate healthcare, primary level, private, pediatrics, infections

## Abstract

In Uganda, >50% of sick children receive treatment from primary level-private health facilities (HF). We assessed the appropriateness of care for common infections in under-five-year-old children and explored perspectives of healthcare workers (HCW) and policymakers on the quality of healthcare at low-level private health facilities (LLPHF) in western Uganda. This was a mixed-methods parallel convergent study. Employing multistage consecutive sampling, we selected 110 HF and observed HCW conduct 777 consultations of children with pneumonia, malaria, diarrhea or neonatal infections. We purposively selected 30 HCW and 8 policymakers for in-depth interviews. Care was considered appropriate if assessment, diagnosis, and treatment were correct. We used univariable and multivariable logistic regression analyses for quantitative data and deductive thematic analysis for qualitative data. The proportion of appropriate care was 11% for pneumonia, 14% for malaria, 8% for diarrhea, and 0% for neonatal infections. Children with danger signs were more likely to receive appropriate care. Children with diarrhea or ability to feed orally were likely to receive inappropriate care. Qualitative data confirmed care given as often inappropriate, due to failure to follow guidelines. Overall, sick children with common infections were inappropriately managed at LLPHF. Technical support and provision of clinical guidelines should be increased to LLPHF.

## 1. Introduction

About half of over five million annual global deaths of children under five years old occur in sub-Saharan Africa [1]. In Uganda, despite some progress, the under-five mortality stands at 46 deaths per 1000 live births. This is above the Sustainable Development Goal (SDG) target 3–25 deaths per 1000 live births [1,2,3]. Like other resource-limited settings, the main causes of deaths among children in Uganda are pneumonia, malaria, diarrheal diseases, and neonatal conditions. These conditions are preventable and reversible if timely appropriate treatment is instituted [4,5,6,7]. Evidence-based low cost interventions to mitigate child deaths offered at community level and primary level health facilities have been described, including preventive interventions and curative interventions, such as antibiotics for pneumonia, zinc therapy and oral rehydration salts (ORS) for diarrhea, early identification of danger signs and referral of newborns for appropriate treatment, prompt diagnosis and treatment of malaria with artemisinin-based combination therapies [8,9,10,11,12]. These interventions can be offered by public and private lower level health facilities, and a recent modelling study at community level demonstrated that they could reduce child mortality significantly [13].

Private health facilities offer an important service, filling the gaps in healthcare delivery in Uganda, with over 50% of sick children being first treated by primary level private health facilities [14]. In low-income countries (LIC), especially in semi-urban and rural areas, private health facilities range from unregistered small drug shops, to registered clinics manned by mid-level health workers, and often only a few have the basic prerequisites to provide good quality health care [15,16,17]. Kruk et al. in their study on variation in quality of primary-care in seven African countries including Uganda showed that private facilities performed better than public facilities [18]. On the other hand, a study done in rural Uganda showed no difference in the quality of care for sick newborns in private practice compared to public facilities [19]. Healthcare is said to be of good quality if it increases the likelihood of desired health outcomes and is consistent with current professional knowledge [20]. Many lower-level private health facilities in rural Uganda are unregistered, often operated by inadequately trained health providers and may not be regulated by the Ministry of Health (MoH) [21,22,23].

The Integrated Management of Childhood Illnesses (IMCI) is a strategy that can be used to deliver interventions for common childhood conditions, improving health worker skills and reducing mortality among children less than five years of age [24,25,26,27]. Managing several conditions in an integrated manner saves time and ensures all conditions are treated without missing or delaying treatment [28]. In Uganda, IMCI was implemented nationally in 1995, and had been rolled out throughout the country by 2003. It was adapted as a pre-service course for health workers, including nurse assistants, and eventually modules for community IMCI and IMCI for private practitioners were introduced [29].

We assessed the appropriateness of care for common infections in under-five year old children with emphasis on pneumonia, malaria, diarrhea, and young infants with possible serious bacterial infections (PSBI) by healthcare workers at low-level private health facilities (LLPHF) in Mbarara District in Western Uganda. We also explored the perceptions of policymakers and healthcare providers on the quality of care provided. We used Integrated Management of Childhood Illnesses (IMCI) guidelines [30] as the reference standard. Appropriateness of care is a measure of the technical quality of healthcare. Together with the assessment of the range of services offered, counselling quality, and promotion of continuity of care, they comprise the process domain of the Donabedian framework for quality of healthcare [31].

## 2. Material and Methods

### 2.1. Research Design

This was a mixed-methods convergent parallel study [32,33] with the quantitative aspect as the dominant part of the study. A cross sectional design was used to collect quantitative data through interviews and observations. We used individual interviews for qualitative data collection. We used a mixed method research design to ensure validity of the study by complementing findings in the quantitative study arm with the qualitative findings. We used the quantitative methods to quantify child consultations that are managed appropriately, while qualitative methods were used to understand how care providers and policymakers perceived the quality of care provided by LLPHF. Data were analyzed separately and merged at interpretation of the results as illustrated in Figure 1.

### 2.2. Study Setting

The study was conducted in Mbarara District in western Uganda, among LLPHF offering pediatrics care, the district health office and at the Uganda Ministry of Health offices. Mbarara is a rural district located 267 km south-west of Kampala, the capital city of Uganda. Further description of the district is summarized in Table 1.

The main health facility is Mbarara Regional Referral Hospital, a tertiary referral facility and teaching hospital for the medical school of Mbarara University. The majority of private health facilities in Mbarara are at health center (HC) III level or lower in the Uganda MOH facility structure [34]. In the context of this study, we defined such health facilities as low-level. The HC IIs provide preventative, promotive, and outpatient curative services, and emergency maternal deliveries. The HC IIIs in addition provide inpatient, maternity, and laboratory services.

Most of the private clinics are run by nurses, midwives, or clinical officers whose highest level of qualification are mostly certificates or diplomas, while a few have degrees. In more rural facilities, the services may even be delivered by individuals with minimal training in health-related fields such as nursing assistants and laboratory assistants [35,36].

The health facilities are registered and regulated by professional bodies to which the facility heads subscribe. The medical doctors are regulated by the Uganda Medical and Dental Practitioners Council (UMDPC), the nurses and midwives by the Uganda Nurses and Midwives Council (UNMC), while the clinical officers are regulated by the Allied Professionals Council (APC).

### 2.3. Sampling and Recruitment

A multistage consecutive sampling technique was employed in the quantitative part, to select facilities, health workers, and caretakers of children below 5 years of age being treated for malaria, pneumonia, or diarrhea, and young infants with possible serious bacterial infections at LLPHF for assessment of appropriateness of care. The sample size was estimated using the formula for single proportions with adjustment for clustering [37,38]. We anticipated that children treated from the same health facility are likely to get similar care, hence the presence of clustering at the health facility level. A sample size of 770 consultations was calculated to identify the proportion of sick children under 5 years of age who got appropriate care for possible serious bacterial infections, malaria, pneumonia, or diarrhea using IMCI guidelines as reference standard [30]. We assumed a 95% confidence interval, precision of 5%, and an arbitrary figure of 50% as the proportion of under-5-year-olds who receive appropriate care for common childhood infections in LLPHF. All private health facilities were screened for eligibility and among the eligible LLPHF a maximum of 10 eligible children per facility whose caretakers were interviewed were enrolled till the desired sample size was met.

In the qualitative part we employed a multi-level purposive intensity sampling to select the 38 key informants. These included eight key policymakers dealing with child health for at least 12 months at the Ministry of Health and the district health office, and 30 healthcare workers (HCW) in private practice managing children for at least 6 months.

#### Selection Criteria

Health facilities were included if they were privately owned, were at a level of health center III or lower, of the MoH health facility classification, and treated children. Health facilities were excluded if the in-charge did not give informed consent, or they were drug shops that simply sold medicines. The HCW were observed only if they were carrying out a consultation for children aged less than 5 years with the following symptoms: fever of any duration, cough, difficult or fast breathing, diarrhea, or a young infant aged less than 2 months, presenting with any symptom or sign of ill health. Only children whose caretakers gave informed consent to participate in the study were observed being managed. Policymakers at the district health office and MoH were included if they worked in the department or unit dealing with children for at least 12 months and HCW who had been managing children in LLPHF in the past 6 months.

### 2.4. Variables

The outcome variable was appropriateness of care for children treated for malaria, pneumonia, or diarrhea and PSBI. The care was assessed against IMCI guidelines and considered appropriate if the process of diagnosis conformed to guidelines, the classification of the child’s illness was correct, the correct medicines were prescribed in appropriate doses and the child was referred when appropriate [30]. A child was said to have pneumonia if she/he had a cough or difficult breathing or fast breathing for age. Diarrhea was present if the child presented with history of passing 3 or more loose motions in 24 h in past 7 days. Malaria was diagnosed if the child had any history of fever, felt hot or had a high temperature and positive malaria test. An infant less than 2 months was said to have a PSBI if the infant was sick in any way; fever, low body temperature, failure to breast feed, difficult breathing etc. Classification of the diseases conditions and treatment were as per IMCI guidelines [30]. Other variables were (1) socio-demographic characteristics of the child and caretaker including age, sex, education level, occupation, socio-economic status, religion, (2) clinical characteristics of the child: presenting symptoms and time since onset of symptoms, (3) health provider characteristics, including qualifications, sex, duration in service, number of health facilities he/she works in, whether providing services in government facility, as well and refresher courses undertaken since the primary qualification, (4) characteristics of the health facility including location (urban or rural) and distance from nearest public health facility.

### 2.5. Quantitative Data Collection

Quantitative data were collected by six research assistants (RAs), all practicing medical doctors previously trained in IMCI. They under-went 3-day training on the protocol, study tools, and a refresher on IMCI. The data capture tool was piloted on six participants. Each consultation was assessed by 2 RAs at different points and working independently, one in the consultation room and the other outside the consultation room. After obtaining informed consent from HCW and care takers of the sick children, the first RA in the consultation room observed HCW during routine consultation for the children aged ˂5 years with fever, cough, diarrhea, or young infants with a suspected serious infection, and recorded the observations on a structured practice observation form. When the HCW had completed the consultation, written the notes, and discharged the patient from the consultation room, the second RA carried out an exit interview with the caretakers of the children, checked the information written and treatment prescribed by the HCW on the patient’s medical form. The second RA then re-examined the patient for any discrepancies and filled the information on the data capture tool. Where there was a discrepancy in the findings and treatment, the second RA then wrote the correct diagnosis and treatment for the patients. At the end of the day’s data collection with the particular HCW, the RAs met to clean the data and discussed the discrepancies and gave feedback to the HCW about their performance in assessing and treating the children seen. This was meant to improve the HCW skills but without biasing the data collection. The data capture tool was modified from the Institute of International Programs at Johns Hopkins University tool for assessing quality of care for sick children and the IMCI health facility assessment tool [39].

#### Statistical Analysis

Quantitative data were analyzed using STATA version 15 (STATA Corp LP, College Station, TX, USA). Descriptive analysis was done for health facilities, health workers, and sick children and their caretakers. To identify factors influencing appropriateness of care, we performed mixed-effects logistic regression models. We included variables as categorical fixed effects nested within fixed facility identifiers and assumed normal distribution of the random effects. We first conducted univariable analyses to estimate crude odds ratios. In order to obtain adjusted odds ratios, variables with *p*-value < 0.1 level in univariable analyses were then included in final multivariable models. Prior to inclusion in final models, multi-collinearity among independent variables was detected using variance inflation factors. We adjusted the final model for ever receiving refresher training by the HCW, given the known interactive influence of this variable on appropriateness of care. In addition to including a design effect in the sample size calculation and limiting the number of consultations per facility to 10, we estimated standard errors to account for clustering of observations within facilities. Variable in the final model with *p* < 0.05 were considered statistically significant. Data are presented as proportions, frequencies, and odds ratios with *p*-values and 95% confidence intervals.

### 2.6. Qualitative Data Collection

Semi-structured interviews were conducted between May and December 2019 by two female (BK and PT) and 1 male (CO) Ugandan RAs with prior training and experience in qualitative research. All the participants were identified with the help of their respective supervisors. To capture the heterogeneity in rural and urban health facilities, a maximum of 10 health workers per county were interviewed. Among the policymakers, one was from the Ministry of Health headquarters while eight were from of the district health management team, which is the decision making body for health sector at the district level. The total sample size of 38 participants was determined through data saturation, or the point when additional interviews did not point to new information [40,41]. All interviewers were fluent in English and Runyankore, the dialect spoken by majority of people in Mbarara. The RAs did not know any of the study participants before conducting the interviews. Prior to study initiation, the RAs were trained on the study protocol, principles of qualitative data collection, and how to conduct high-quality interviews, interview translation, and transcription. An interview guide consisting of open-ended questions was specifically created for this study to ensure consistent focus on the views on appropriateness of care, reasons explaining the current situation and how to improve it. All the interviews were carried out in a private room at the respective workplaces of the interviewees, at a time convenient to the respondents. Each interview lasted 30–40 min and was audio-recorded with the participant’s permission. All the interviews were carried out in English as this was the participants’ preference. All interviews were transcribed verbatim by the interviewer, based on audio recordings. JMA read the transcripts line by line within 48 h of transcript completion and provided feedback to the RAs to continuously improve their interview skills throughout the data collection period. This ensured consistency in quality and content across all interviews and served as a means to monitor for data saturation.

To ensure anonymity, the respondents were de-identified using study numbers. For policymakers where gender and employment position lead to disclosure of participant’s identity we opted not to include the gender of the respondents in the quotes in order to further de-identify them. To ensure confidentiality of study data, transcripts and the voice recordings were transferred to a password protected locked computer only accessed by JMA who shared a password protected folder with JO for coding. Printed documents such as signed consent forms were kept in a locked cabinet accessible only to JMA.

#### Qualitative Data Analysis

Data were analyzed thematically employing a deductive approach [42]. After reviewing each transcript several times, JMA and JO independently developed the initial set of codes. JMA and JO then discussed and compared their codebooks and through consensus the codes were revised to create a final code list. The codes were grouped together into subcategories and categories pertinent to the research question. The preliminary list of subcategories and categories were shared and discussed with the other authors to come up with the final categorization.

## 3. Results

We visited 120 private clinics, 110 of which fulfilled the enrolment criteria, and observed 777 medical consultations, an average of seven consultations per health facility. Over 70% of the consultations took place in facilities at the level of HCII or below. Only 10% of the consultations took place in facilities headed by medical doctors. The median age of the health workers who carried out the consultations was 27 years (IQR 24, 33) and median period in service was 3.4 years (IQR 2.0, 9.6). The characteristics of the health facilities in which the study took place and health workers are shown in Table 2.

The majority of the consultations were carried out by health workers who had not had any refresher training in IMCI nor other stand-alone refresher trainings in the management of common childhood illnesses in the past 6 years as shown in Figure 2.

The mean age of the children was 24 months (SD 16.0). Fever was present in 549 (71%), cough in 469 (60%), while 146 (19%) children had danger signs. The clinical presentation of the children is shown in Figure 3 while their demographic characteristics, as well as those of their caretakers are in Table 3.

Most of the caretakers were peasants with a median monthly income of Uganda shillings 100,000 (IQR 30,000, 300,000), approximately 27 USD and 47% had attained at most primary school education.

### 3.1. Management of the Sick Children by the Health Workers

The management was said to be appropriate if all identification, classification, and treatments were done correctly. As shown in Table 4, the clinical conditions were correctly identified in the majority of the consultations but wrongly treated. Among the 18 children deemed to have possible serious bacterial infections, only one was identified by the health workers as such. All conditions were inappropriately managed in over 80% of the consultations.

### 3.2. Factors Associated with Child Receiving Appropriate Care

Table 5 shows the regression analysis for factors associated with appropriate care. Several factors including caretaker age >25 years, rural location of the health facility, and clinical conditions that the child presented with were associated with the appropriateness of care at univariable analysis. However, after controlling for interaction and adjusting for confounding in the multivariable analysis, only the presenting clinical condition was associated with appropriateness of care received. The care was less likely to be appropriate for children with diarrhea (OR 0.29, 95% CI: 0.11–0.76, *p* = 0.012), as well as those able to feed orally (OR 0.07, 95% CI: 0.03–0.13, *p* < 0.001). The care was likely to be appropriate among more severely ill children with danger signs including those who were vomiting everything (OR 10.1, 95% CI: 4.41–22.1, *p* < 0.001), had seizures (OR 7.54, 95% CI: 2.97–19.1, *p* < 0.001), or were lethargic or had impaired consciousness (OR 4.42, 95% CI: 1.03–19.1, *p* = 0.046).

### 3.3. Perspectives of Policymakers and Health Workers on Quality of Care Offered by LLPHF

During the same period, we carried out individual interviews with 30 healthcare workers and 13 policymakers to find out what they thought about the care given by LLPHFs. The main finding was health care being inappropriate. This is described in four subcategories and these results complement the quantitative results that show high level of inappropriate health care.

(1)Referral of patients with severe conditions may not be timely(2)Inadequate infrastructure exacerbates inappropriate healthcare provision(3)The care is not patient centered but rather is driven by desire to make money or please the caretaker(4)Existing guidelines are not followed, leading to mismanagement patients

### 3.4. Referral of Patients with Severe Conditions May Not Be Timely

Participants confirmed that LLPHF mainly handle children with uncomplicated illnesses and should refer the severe cases, however some facilities delay referrals and in the long run mismanage such patients.


*“Most of the low-level private clinics treat minor conditions, like simple coughs, malaria, pneumonias, they should not manage serious illnesses, as these should be referred to higher levels. The challenge is some facilities wait and only refer the child in very late stages.”*
(Male policymaker, DHO)


*“Here we mostly handle outpatient clients and a few in patients especially when conditions don’t require us to refer. To give examples we handle simple malaria, we handle respiratory infections including pneumonia because our facility is categorized as health center III.”*
(Male nurse, rural clinic)

### 3.5. Inadequate Infrastructure Exacerbate Inappropriate Healthcare Provision

Most of the LLPHF have minimum infrastructure and inadequately trained staff. Many do not have proper laboratories and rarely carry out investigations, which often make the provision of appropriate care difficult.


*“It is expensive paying an enrolled nurse….so they prefer to hire nursing aids, who you can pay little money. You find expired reagents and untrained lab personnel simply picked because they are relatives to the owner of the clinic. The fridges are not functional, instead, they just put water in the bucket and put reagents and they refer to that as a fridge. So how do you expect to find a correct result….?”*
(Female policymaker from DHO)


*“Most of us don’t carry out investigations so when the patients come with cough you give septrin (Cotrimoxazole), amoxicillin, plus Panadol (paracetamol) if the fever persists that is when the child may be taken for testing somewhere…..”*
(Female nurse, rural clinic)

The care is not patient-centered but is rather driven by the desire to make money or please the caretaker. Often patients do not receive appropriate care because they are given unnecessary medications even for simple illnesses. This is said to be driven by the desire to make money and in some instances the unprofessional behaviors stem from the urge to satisfy the caretaker instead of the condition of the child.


*“People want money; some may provide unnecessary treatment to make the treatment list very big and then get more money. I witnessed that behavior several times in one of the clinics I previously worked for.”*
(Male clinical officer, rural clinic)


*“There is irrational use of antibiotics in private clinics partly because it is always difficult to explain to the mother that the child’s simple cough does not need an antibiotic…. Sometimes you give that higher antibiotic to impress the mother but of course knowing that you are not doing the right thing. Unfortunately the children somehow have been mismanaged, the facilities have learnt the fact that if you don’t inject you don’t get the money.”*
(Male doctor, urban clinic)

### 3.6. Existing Guidelines Are Not Followed, Leading to Mismanagement of Patients

Participants noted that even if national guidelines for managing childhood illnesses exist, they are not utilized by all LLPHF. This is either because the health workers at the LLPHF often do not have access to the guidelines, they do not know about their existence, and have not been trained on their use, or they just do not bother to follow these guidelines. In the long run, there is mismanagement of children and misuse of medicines.


*“We sometimes attend conferences where they elaborate the new clinical guidelines that come up. Some of us follow the guidelines but many others do not. When they see someone has a fever, has a complication they start off by pumping the drugs.”*
(Male doctor, urban clinic)


*“When we are assessing these children we use the knowledge we acquired from school but we have not been given any guidelines to use; that is the challenge.”*
(Male clinical officer, rural clinic)

Some other participants, however, intimated that the services varied from clinic to clinic and that because of competition, the LLPHF endeavor to provide fair services, otherwise they lose their clientele.


*“The services are fair….if the child is taken to the facility and is not well managed in the next one day will be moved to the next health facility for better services….we are in town and they can easily get to a number of facilities around.”*
(Male doctor, urban clinic).

## 4. Discussion

We assessed appropriateness of care and associated factors for children aged ˂5 years presenting with pneumonia, malaria, diarrhea, or possible serious infections at LLPHF in Mbarara, and explored how the policymakers and healthcare providers perceived the quality of care provided. We found that majority of children presenting with these common childhood infections were not managed appropriately. While pneumonia, diarrhea, and malaria were diagnosed correctly in most of the consultations, often the given treatment was wrong. Almost all infants with a possible serious bacterial infection were wrongly diagnosed and treated. Children who had danger signs, especially vomiting everything, seizures, and lethargy or impaired consciousness were more likely to receive appropriate medical care, while those who were able to feed orally were more likely to receive inappropriate care. The qualitative results complemented the quantitative results and showed that policymakers and the health workers perceived the healthcare provided by LLPHF as inappropriate because guidelines were not being followed and the care was often driven by the desire to make money or please the caretaker than the clinical needs of the patient.

While pneumonia, diarrhea, and malaria were identified correctly according to IMCI guidelines, the wrong treatment was often administered either by giving unnecessary medications, the wrong medicine, or omitting important medications, which translated into inappropriate care of the conditions. This is contrary to the standards set by the WHO in 2018, which require that all sick infants, children with cough, diarrhea, or fever are thoroughly assessed, classified, and given correct medications for the condition [43]. Inappropriate care for these common childhood infections has been described by other researchers in Uganda and other LMICs. Mbonye et al. in their research among private clinics and drug shops found that children with normal respiratory rates received unnecessary antibiotics for pneumonia [44], while Kjaergaad et al. found inappropriate prescription rate of antibiotics for viral upper respiratory tract infections of 23–68% in Uganda and other countries across the globe [45]. Other studies carried out in private and public health facilities in Uganda and other African countries found that 7–24% children testing positive for malaria never received the correct malaria medicine [46,47]. In a study from Nigeria, over 85% of children with watery diarrhea were given unnecessary antibiotics, while in a study from Ethiopia, the assessment, as well as treatment of children for cough, fever, and diarrhea in clinics was of poor quality [48,49]. In our study, it is probable that the health workers were able to diagnose the conditions using the knowledge they had from their professional training since the presentation of the conditions remains the same. On the other hand, treatment of diseases is updated often as better tools become available and new evidence emerges, necessitating policy change. If the dissemination of new policies is not carried out in tandem with their introduction and refresher training, the health workers may continue giving the old treatments because they have not acquired the new knowledge. In addition, even if the new knowledge is passed on, it would take some time for people to change the behavior they are used to. This therefore necessitates supportive supervision or a closer follow up by the policy implementation team at the district. The LLPHF are often left out in refresher training and in addition often do not have access to current treatment guidelines [15,50,51,52]. The fact that young infants with possible serious bacterial infections were wrongly diagnosed and consequently inappropriately treated could be explained by the limited knowledge of diseases affecting young infants, an area needing specialists, yet the health workers at LLPHF are mostly of a lower cadre. A community inquiry carried out in the same region and central Uganda as well as Mali also identified mismanagement as cause of deaths among newborns [53]. A recent research policy working paper by the World Bank highlighted the lack of clinical knowledge as a major barrier to quality care provided by health facilities in 10 African countries including Uganda [54].

Children without danger signs were more likely to receive inappropriate medical care while those with danger signs were more likely to receive appropriate care. Most patients who present to these LLPHF have mild illness, however they are administered with medication that should be reserved for severe cases. They, for example, do not need antibiotics for a mild cough, nor need injections for nonsevere pneumonia or uncomplicated malaria. They are able to swallow and should be given oral medications where necessary, not injections. For children with danger signs, this may be justifiable thus the treatment will be appropriate. This injudicious prescription of medications in private practice has been described by other researchers, for example, in Uganda and in India [52,55,56]. Possible explanation to this phenomenon includes lack of knowledge among the providers in LLPHF, lack of clinical guidelines, or a drive for financial incentives. Indeed, in our study, the policymakers perceived the care as driven by financial incentives, wanting to please care takers rather than the patient needs, and failure for providers to follow clinical guidelines. This phenomenon of unprofessionalism has been observed by other researchers in other areas [55]. A study done in Kenyan hospitals demonstrated that availability of clinical guidelines was associated with appropriate prescription of antimicrobials [57]. While this Kenyan study was carried out in public hospitals, it can still be extrapolated to LLPHF that offer mainly outpatient care.

As a limitation to this study, observing health workers perform could have introduced the Hawthorne effect. This refers to a tendency of people to modify their behavior when being observed. In this instance, the healthcare workers would act more carefully and perform better than they usually do. This effect was minimized by the health workers being observed on more than one child. This increased rapport between them and the RAs, thus enabling the healthcare workers to become comfortable and act normal. Another limitation was that we did not collect data on the quality of the refresher training that the HCW received. This could have probably helped us to understand why refresher training did not result in better performance. The mixed concurrent multi-level design allowed data collection using different methods and thus strengthened the validity of the study while interviewing a wide range of participants at different levels facilitated getting views of the policymakers, as well as care providers.

## 5. Conclusions

The care provided for common pediatric infections by LLPHF in Mbarara District in Western Uganda was often inappropriate. While the clinical conditions were often identified correctly, wrong treatment was given. Serious infections among young infants were very often missed and wrongly treated. Not having a danger sign was associated with receiving inappropriate medical care.

We recommend close technical support to LLPHF through mentorship either physically or using telemedicine. We also recommend that IMCI training is revitalized by the Ministry of Health for health workers in LLPHF and free digital or printed clinical guidelines are provided and their use encouraged.

## Figures and Tables

**Figure 1 ijerph-18-07742-f001:**
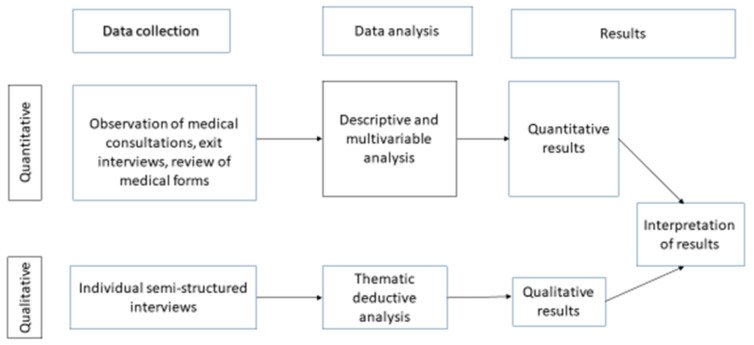
Graphic representation of the convergent parallel study design.

**Figure 2 ijerph-18-07742-f002:**
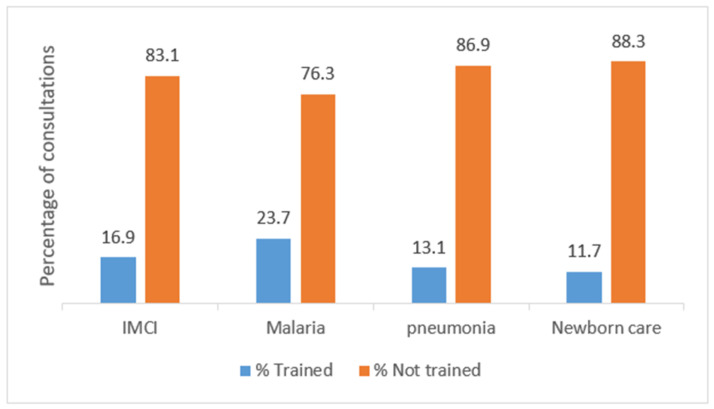
Percentage of consultations in low-level private health facilities in Mbarara, carried out by healthcare workers who had or not received refresher training in management of common childhood infections in the past 6 years.

**Figure 3 ijerph-18-07742-f003:**
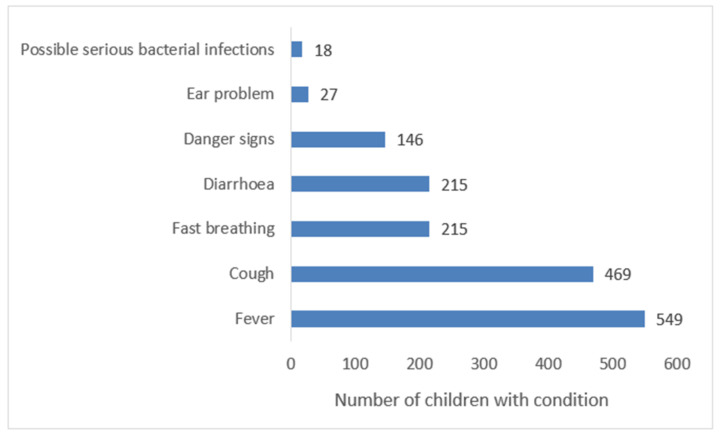
Clinical problems at presentation among children seeking medical care in low level private health facilities in Mbarara Uganda.

**Table 1 ijerph-18-07742-t001:** Some characteristics of Mbarara District.

Characteristic	Number
Population (inhabitants)	470,000
Population density/km	99
Administrative divisions	
Counties	3
Sub-counties	16
Parishes	83
Villages	742
Health facilities	
Hospitals	7
Health centers	49
Registered private health facilities (in 2017)	124

**Table 2 ijerph-18-07742-t002:** Number of consultations by health facility and health worker characteristics.

Characteristic	Number of Consultations *n* (%)
Health facilities	
Level of health facility (N = 777)	
Below HCII	232 (30)
HCII	337 (43)
HCIII	208 (27)
Location of health facility (N = 777)	
Urban	311 (40)
Rural	466 (60)
Professional qualification of the health facility head (N = 776)	
Doctor	79 (10)
Clinical officer	310 (40)
Nurse/midwife	292 (38)
Nursing assistant	92 (12)
Laboratory technician	3 (0.4)
Health facility licensing body (N = 776)	
Uganda Medical and Dental Practitioners Council	92 (12)
Allied Health Professionals Council	377 (49)
Uganda Nurses and Midwives Council	108 (14)
Unregistered	199 (26)
Healthcare workers, (N = 776)	
Males	353 (45.5)
Females	423 (54.5)
Working only in private health facilities	704 (90.7)
Working in two or more health facilities	229 (29.5)

**Table 3 ijerph-18-07742-t003:** Characteristics of the children and their caretakers.

Characteristics	Summary Statistics
Children	
Sex (N = 777)	
Male, *n* (%)	376 (48.4)
Female, *n* (%)	401 (51.6)
Mean age in months, (SD)	24.0 (16.0)
Mean weight in kg, (SD)	11.1 (3.7)
Mean height/length in cm, (SD)	75.7 (20.9)
Temperature ≥37.5 °C, *n* (%)	390 (60.3)
Caretakers	
Relationship with the sick child, *n* (%)	
Biological parent	695 (89.4)
Sibling	2 (0.3)
Others (grandparents, aunts, uncles)	80 (10.3)
Sex, *n* (%)	
Male	130 (16.7)
Female	647 (83.3)
Age, mean (SD)	30 (8.0)
Residence, *n* (%)	
Rural	481 (61.9)
Urban	296 (38.1)
Level of education, *n* (%)	
None	72 (9.3)
Primary	366 (47.1)
Secondary	230 (29.6)
Tertiary	109 (14.0)
Marital status, *n* (%)	
Single/divorced/widowed	160 (20.6)
Married/living with a partner	617 (79.4)
Median monthly income in Uganda shillings, (IQR)	100,000 (30,000, 300,000)

**Table 4 ijerph-18-07742-t004:** Diagnosis, classification, and treatment of sick children by health workers at LLPHF.

	Condition, *n* (%)			
	Pneumonia	Diarrhea	Malaria	Possible Serious Bacterial Infection
Correct identification				
No	62 (15)	35 (16)	142 (27)	17 (94)
Yes	360 (85)	183 (84)	390 (73)	1 (5.6)
Correct classification				
No	44 (10)	57 (26)	350 (66)	17 (94)
Yes	378 (90)	161 (74)	182 (34)	1 (5.6)
Correct treatment				
No	367 (87)	123 (56)	446 (84)	12 (67)
Yes	55 (13)	95 (44)	86 (16)	6 (33)
Appropriately managed overall				
No	393 (89)	204 (92)	457 (86)	18 (100)
Yes	49 (11)	18 (8.1)	75 (14)	0 (0)

**Table 5 ijerph-18-07742-t005:** Factors associated with appropriateness of care.

	% Appropriate Care	Univariable Analysis		Multivariable Analysis	
Characteristic	*n*/N (%)	cOR (95%CI)	*p*-Value	aOR (95%CI)	*p*-Value
Caretaker’s age					
<25 years	7/154 (4.6)	Ref		Ref	
≥25 years	71/614 (11.6)	2.75 (1.24–6.10)	0.013	1.95 (0.71–5.37)	0.194
County of H/F location					
Rwampara	9/177 (5.1)	Ref			
Kashari	40/238 (16.8)	3.77 (1.78–8.00)	0.001		
Municipality	29/326 (8.0)	1.63 (0.75–3.51)	0.216		
Location					
Urban	22/311 (7.1)	Ref		Ref	
Rural	56/466 (12.0)	1.79 (1.07–3.00)	0.026	1.09 (0.62–1.93)	0.756
Time lag to nearest public H/Facility					
≤15 min	44/550 (8.0)	Ref		Ref	
>15 min	34/226 (15.0)	2.04 (1.26–3.28)	0.004	2.50 (1.23–5.10)	0.012
Residence of caretaker					
Rural	58/481 (12.1)	Ref		Ref	
Urban	20/296 (6.8)	0.53 (0.31–0.90)	0.018	0.58 (0.23–1.45)	0.247
Temperature of child					
<37.5 °C	17/257 (6.6)	Ref			
≥37.5 °C	52/390 (13.3)	2.17 (1.23–3.85)	0.008		
Presence of fast breathing					
No	38/562 (6.8)	Ref		Ref	
Yes	40/215 (18.6)	3.15 (1.96–5.07)	<0.001	1.77 (0.95–3.12)	0.072
Presence of diarrhea					
No	62/561 (11.1)	Ref		Ref	
Yes	16/216 (7.4)	0.65 (0.36–1.15)	0.137	0.29 (0.11–0.76)	0.012
Child able to drink					
No	43/84 (51.2)	Ref		Ref	
Yes	35/693 (5.1)	0.05 (0.03–0.09)	<0.001	0.07 (0.03–0.13)	<0.001
Child vomiting everything					
No	50/717 (7.0)	Ref		Ref	
Yes	28/58 (48.3)	12.45 (6.90–22.46)	<0.001	10.1 (4.41–22.1)	<0.001
History of convulsions					
No	57/741 (7.7)	Ref		Ref	
Yes	21/34 (61.8)	19.38 (9.22–40.73)	<0.001	7.54 (2.97–19.1)	<0.001
Level of consciousness					
Awake	69/763 (9.0)	Ref		Ref	
Lethargic or unconscious	9/14 (64.3)	18.10 (5.90–55.53)	<0.001	4.42 (1.03–19.1)	0.046
Neck rigidity					
No	76/773 (9.8)	Ref			
Yes	2/4 (50.0)	9.17 (1.27–66.04)	0.028		
Chest in drawing					
No	55/664 (8.3)	Ref			
Yes	23/113 (20.4)	2.82 (1.66–4.83)	<0.001		
Presence of stridor					
No	72/753 (9.6)	Ref			
Yes	6/24 (25.0)	3.15 (1.21–8.20)	0.018		
Sunken eyes/signs of dehydration					
No	66/712 (9.3)	Ref		Ref	
Yes	12/65 (18.5)	2.21 (1.13–4.36)	0.021	2.07 (0.67–6.33)	0.204
Age of Health worker					
<25 years	13/202 (6.4)	Ref			
25–40 years	57/502 (11.4)	1.86 (1.00–3.48)	0.052		
>40 years	8/72 (11.1)	1.82 (0.72–4.58)	0.206		
Trained on management of malaria					
No	55/592 (9.3)	Ref			
Yes	23/184 (12.5)	1.39 (0.83–2.34)	0.208		
Trained on management of pneumonia					
No	62/675 (9.2)	Ref			
Yes	16/102 (15.7)	1.84 (1.02–3.33)	0.044		
Trained on management of bacterial infections					
No	64/689 (9.3)	Ref			
Yes	14/88 (15.9)	1.85 (0.99–3.46)	0.055		
Trained on IMCI guidelines					
No	64/646 (9.9)	Ref			
Yes	14/131 (10.7)	1.09 (0.59–2.01)	0.787		
Ever received refresher training					
No	51/555 (9.2)	Ref		Ref	
Yes	27/222 (12.2)	1.37 (0.83–2.24)	0.149	1.50 (0.77–2.94)	0.237
Health worker professional qualification					
Nurse/midwife	46/460 (10.0)	Ref			
Doctor/clinical officer	18/191 (4.2)	0.94 (0.54–1.66)	0.822		
Nursing assistant	13/120 (16.7)	1.09 (0.57–2.10)	0.788		
Others	1/6 (1.3)	1.80 (0.21–15.7)	0.595		

## Data Availability

The data that support the findings of this study are available from Juliet Mwanga-Amumpaire upon request.

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
