# Peer review of "Appropriateness of Care for Common Childhood Infections at Low-Level Private Health Facilities in a Rural District in Western Uganda"

_ijerph, 2021, doi:10.3390/ijerph18157742_

Round 1

Reviewer 1 Report

This paper reports the findings regarding the level of appropriateness of care given to kids for common diseases in Western Uganda. The study is very extensive and the manuscript is very well-written and easy to follow. I believe this work and hence the paper is quite valuable in terms of bringing the current inappropriateness of care given to children to readers' attention. Even though the study was performed in Western Uganda, I believe the findings can be easily generalised to a wider region with similar socio-economic conditions.

While the authors have done a great job in explaining the current situation, I think the paper can be improved if they can add analysis for different levels of qualification. Even the number of doctors involved in this study corresponds to 10% of HCWs, I would be keen to learn the level of appropriateness for each qualification level especially for doctors.

In addition, I found it quite interesting and puzzling that getting trained for certain diseases was not a factor. Could this be related to the quality/content of training, time since the training, or some other factors? I think it is valuable to define what being trained means and the authors should elaborate more on this finding.

Some minor comments:

Methods: Please include an ethical approval statement. 
Figure 1: 'Qualitative' was split into two lines
Figure 2: The caption needs revision. Make it clear that these are the percentages for those who were involved in the consultations in your study.
Line 194: HCW was node defined.
Line 197: p<0.005 >> p<0.05
Line 207: DHMT is not used again, so no need to abbreviate. Please check other abbreviations as well and spell out if they are used only a few times.
Line 256: I personally do not find it correct to use plus/minus for SD.
Line 283: able feed >> able to feed
Line 285: space needed between % and CI.
Line 303: period is missing at the end of the sentence.
Line 324: Italicise the sentence.
Line 432: two uses of 'however'

Reviewer 2 Report

The study was a mixed-methods convergent parallel study with the aim of assessing the appropriateness of care for common infections in under-five-year-old children, and explore perspectives of healthcare workers and policy-makers on the quality of healthcare at low-level-private health facilities in western Uganda.

The manuscript is interesting, well designed and well written.  I have very few suggestions:

  1. The introduction has an adequate length. However, the third paragraph should be moved to the end of the Introduction and the objective be more clear (at the end of the introduction).
  2. The methods are appropriated and well explained. I suggest to include in the methods section that the child caretakers also gave consent to the study.
  3. I suggest explaining for the reader what the Hawthorn effect is. Other than that, which are the limitation of the study?

Reviewer 3 Report

Thank you for the opportunity to review the manuscript titles “Appropriateness of care for common childhood infections at low-level private health facilities in a rural district in Western Uganda.”  This is a well designed study and a well written manuscript.  I only have a few minor recommendations for the authors.

Please change “data was” to “data were” and “data is” to “data are” throughout the manuscript.

There is a typo in section 2.5.1.  The manuscript currently says p<0.005 instead of p<0.05

It would be helpful to have a section specific for inclusion/exclusion criteria.
